

# Differential bleaching of quartz and feldspar luminescence signals under high turbidity conditions

Jürgen Mey[1], Wolfgang Schwanghart[1], Anna-Maartje de Boer[2], Tony Reimann[2,†]

[1]Institute of Environmental Sciences and Geography, University of Potsdam, Germany
[2]Soil Geography and Landscape group & Netherlands Centre for Luminescence Dating, Wageningen University, Wageningen, The Netherlands
[†]now at Geographic Institute, Geomorphology & Geochronology, Cologne, Germany, University of Cologne, Germany

*Correspondence to:* Jürgen Mey (juemey@uni-potsdam.de)

## Abstract

Sediment burial dating using optically stimulated luminescence (OSL) is a well-established tool in geochronology. An important but often inapplicable requirement for its successful use is that the OSL signal is sufficiently reset prior to deposition. However, subaqueous bleaching conditions during fluvial transport are vastly understudied, for example the effect of turbidity and sediment mixing on luminescence bleaching rates is only poorly established. The possibility that slow bleaching rates may dominate in certain transport conditions led to the concept that OSL could be used to derive sediment transport histories. The feasibility of this concept is still to be demonstrated and experimental setups to be tested. Our contribution to this scientific challenge involves subaquatic bleaching experiments, in which we suspend saturated coastal sand of Miocene age in a circular flume and illuminate it for discrete time intervals with natural light. We record the in-situ energy flux density received by the suspended grains in the UV-NIR frequency range by using a broadband spectrometer with a submersible probe. Our analysis includes pre-profiling of each sample following a polymineral multiple signal (PMS) protocol. Using the PMS, the quartz dominated blue stimulated luminescence signal at 125 °C (BSL-125) decays slower than the K-feldspar dominated infrared stimulated luminescence signal at 25 °C (IR-25) even under subaerial conditions. The BSL-125 from purified quartz shows the opposite behaviour, which renders the PMS unreliable in our case. We find a negative correlation between suspended sediment concentration and bleaching rate for all the measured signals. For outdoor bleaching experiments we propose to relate the measured luminescence dose to the cumulative received irradiance rather than to the bleaching time. Increases in the sediment concentration lead to a stronger attenuation of the UV/blue compared to the red/NIR wavelength. This attenuation thereby follows an exponential decay that is controlled by the sediment concentration and a wavelength-dependent decay constant, λ. As such λ could potentially be used in numerical models of luminescence signal resetting in turbid suspensions.



## 1.    Introduction

The natural luminescence of quartz and feldspar is widely used in geochronology to determine burial ages, i.e., the time of sediment deposition and coverage (Huntley et al., 1985; Roberts and Lian, 2015). Geochronological methods that exploit luminescence thereby rely on the ability of quartz and feldspar to trap charges from the naturally occurring ionizing irradiation while buried. The rate, at which the minerals trap charges is governed by the concentration and type of

radioactive isotopes (mainly $^{40}$K) and radioisotopes within the U and Th decay chains in the surrounding sediment and the type of charge-trapping anisotropies in the crystal lattice. Upon exposure to sun light or to an artificial light source, the charges recombine and the stored energy is released, a process referred to as bleaching or zeroing. The bleaching is accompanied by the emission of photons, whose amount (light intensity) is proportional to the charge-acquisition time,

hence the duration of burial. Since the number of traps is finite, the grain can attain a state of "saturation". Upon saturation, no more charges can be acquired and the "luminescence clock" stops ticking.

A main assumption of this dating method is that the luminescence signal of the sediment grains is reset during transport or during the residence at the surface before burial, i.e., no luminescence from

previous irradiation episodes is inherited. However, many geomorphic processes such as mass-wasting processes or fluvial transport may lead to very brief exposures of the sediments to sunlight (Rittenour, 2008; Fuchs and Lang, 2009). In fluvial settings, for example, individual grains often experience only limited bleaching prior to deposition, which implies that their ages can be overestimated (Wallinga, 2002). This remnant luminescence signal that is also commonly observed

in modern alluvium (Murray et al., 1995; Jain et al., 2004; McGuire and Rhodes, 2015a), has been attributed to limited bleaching due to the attenuation of light caused by high suspended sediment loads and turbidity (Berger, 1990; Davies-Colley and Nagels, 2008). For river systems dominated by sediment recycling the fraction of well-bleached grains in a sample has further been reported to increase with downstream transport distance (Porat et al., 2001; Jain et al., 2004; McGuire and

Rhodes, 2015b; Guyez et al. 2022). Albeit a limitation to luminescence dating of fluvial deposits, an increase of the amount of bleaching with transport distance would bear the possibility to offer insights into fluvial transport conditions in terms of suspended sediment concentration and/or particle velocity (McGuire and Rhodes, 2015a, b; Gray and Mahan, 2015; Gray et al., 2017). However, there still exists a knowledge gap concerning the accurate quantification of such

bleaching rates, which restrains the development of luminescence as a sediment tracer (Gray et al., 2019).



Previous studies suggested to use the IRSL in feldspars for tracing subaqueous sediment transport (e.g.,McGuire and Rhodes, 2015a), because this signal appears to bleach more slowly than the OSL from quartz (Godfrey-Smith et al., 1988). Following theoretical considerations, however, feldspar

may not be the mineral of choice for subaqueous tracing if the turbidity reaches a certain threshold. First, the bleaching efficiency of quartz and feldspar varies for different parts of the light spectrum (Spooner 1994a,b) as the ultraviolet (UV) and infrared (IR) spectrum have different subaqueous transmission characteristics (Wallinga, 2002). While the bleaching efficiency of quartz OSL in the blue part of the UV/VIS spectrum (~400 nm) is approximately 10 times higher than feldspar IRSL,

the latter bleaches significantly more efficient than quartz OSL for wavelengths exceeding 500 nm (Spooner 1994a,b; Wallinga, 2002, his Fig. 4). The subaqueous transmission characteristic is controlled by the concentration and type of dissolved and/or suspended materials in the water (e.g., Berger and Luternauer, 1987; Ditlefsen, 1992; Wallinga, 2002; Sanderson et al., 2007). Secondly, while in pure water the attenuation length of UV is about two orders of magnitude larger than that

of IR (Pope and Fry, 1997), this difference reduces for clear ocean water (Bradner et al., 1992) and eventually inverts in turbid conditions (Gallegos et al., 1990). This implies that high suspended sediment concentrations ($C_s$) and turbidity not only reduce the intensity of light received by the grains but also alter the shape of the light spectrum. The attenuating effect of an increase in turbidity may be disproportionately larger for the UV than for the IR wavelengths. The combined

effects of water and sediment on light spectra and different bleaching behavior of quartz and feldspar imply a threshold in $C_s$/turbidity, beyond which the UV/blue stimulated quartz OSL bleaches slower than the feldspar IRSL. Establishing such threshold may eventually help to infer paleo-turbidities from differential bleaching of quartz and feldspar in fluvial deposits or other depositional environments with a significant contribution of subaquatic transport prior to burial (e.g.

glacial fluvial, shallow marine etc.). To test this hypothesis and to quantify bleaching rates in sediment-laden water we conducted flume experiments with saturated coastal sand of Miocene age and registered the bleaching response of OSL and IRSL as a function of the $C_s$/turbidity and illumination time.

## 2.   Data and Methods

### 2.1.   Bleaching experiment and measurement of light spectra

The experimental character of this study required a large saturated sample containing quartz and feldspar that enables us to measure OSL and IRSL within the same measurement sequence. We collected coastal sand of Miocene age from "Grube Gotthold" (51.5038° N, 13.5310° E), an abandoned open-pit coal mine near the village of Hohenleipisch in southern Brandenburg, Germany

(Fig. 1, Stackebrandt and Franke, 2015). We took two samples (GG1 & GG2) from a homogeneous,



weakly lithified and quartz-dominated horizon underlying a prominent bed of lignite. In addition, we sampled a second sand layer on top of the lignite and directly below the surface (GG3). Sampling was conducted under red light conditions during the night using a shovel and a light-proof cover. To obtain saturated material, we removed the outer ~10 cm of the sediment before sampling.

All experiments were conducted with material from GG1, which is a well sorted fine sand (0.063–0.2 mm) composed mainly of quartz and a few feldspar grains. A dark brown coating is attached to some of the grains that we tentatively classify as organic material (Fig. 1c).

Our experimental setup uses natural light in an outdoor lab, which is more realistic then the use of a solar simulator (Höhnle SOL2 for example) in an indoor environment in terms of bleaching

spectrum and bleaching light intensity. We induced circular flow within a glass cylinder filled with 1.5 l of clear tab water using a magnetic stirrer and added the sample under red-light conditions in a light-proof tent (Fig. 2). Next, the flume containing the sample was placed outside the tent for a certain time interval. During the illumination period, the turn rate of the magnetic stirrer was adjusted so that all grains remained in suspension. Simultaneously, we measured the in-situ

intensity and wavelength of light received by the grains in the UV–NIR interval using an Ocean Insight Flame-T miniature spectrometer and a submersible probe. This setup enabled us to account for variable illumination due to clouds and the change of the sun's elevation. The probe was lowered into the suspension to ~3 cm above the bottom, where sediment concentration was high and interaction with the magnet of the stirrer could be avoided. The water depth was ~12 cm. The

limited bending radius of the light guide connecting the probe with the spectrometer required the probe to look sideways towards the centre of the flume. The probe itself posed an obstacle to the flow that induced turbulence and caused a mixing of the grains throughout the water column. To ensure that light entered the flume exclusively from the top, we wrapped the cylinder in a non-transparent foil with a diffuse reflecting side directed inwards. After each time interval we returned

the flume back into the dark tent and extracted a small subsample for luminescence analysis. We illuminated the samples for time intervals of 5, 10, 20, 40, 80, 160 and 320 minutes and with high suspended sediment concentrations of 33, 66 and 100 g l$^{-1}$ spanning a wide range of observed peak values during floods (e.g., Müller and Förstner, 1968; Hicks et al., 2000; Lenzi and Marchi, 2000; Wulf et al., 2012; Andermann et al., 2012). With a sediment density of 2.65 g cm$^{-3}$ these values

correspond to 1.25, 2.5 and 3.8 vol. % of sediment, respectively. Further, we performed a single subaerial bleaching run with a sample distributed as a monolayer with a thickness of one grain on a flat, reflective surface and illuminated it for 0.1, 1, 5, 10, 20, 40, 80 and 160 minutes while measuring the solar irradiance. The experiments were conducted between 5 April 2020 and 9 May 2021 between noon and 6 pm (Table 1).





Before conducting the bleaching experiments, we determined the wavelength-dependent
discrimination of light in turbid suspensions with sediment loads of 0–100 g l$^{-1}$ in 10 g increments.
These measurements aimed at verifying that the width of the spectrometer's optical slit was suitable
for our purpose. As we operate with high sediment concentrations, the use of a wider slit permits a
higher throughput of photons, thereby increasing the signal-to-noise ratio of the measurements and

hence the sensitivity of the instrument. However, higher sensitivity is achieved at the cost of
reduced spectral resolution. The spectrometer assembly was calibrated against a standardized target
by the manufacturer providing the transfer function to convert the measured intensities, i.e. count
rates, to solar irradiance in $\mu Wcm^{-2}nm^{-1}$. Note that we use the terms irradiance, energy flux density
and light intensity synonymously throughout this manuscript.

**2.2.    Sample preparation and determination of dose reduction**

We adopted a polymineral multiple-signal (PMS) approach that measures different luminescence
signals from single polymineral coarse-grain aliquots (Reimann et al. 2015). The approach enabled
us to measure the bleaching data in a labor-efficient manner and to obtain multiple luminescence
signals from the same sub-set of grains. Sample preparation and measurements were conducted in

the Netherlands Centre for Luminescence dating (NCL) at Wageningen University and Research
(WUR) in the Netherlands.

Altogether, 38 samples (see Table 2) were first dried and sieved to a grain-size of 180 to 250 µm.
Afterwards the samples were purified using 10 % HCL and 10 % $H_2O_2$ to remove carbonates and
organic matter, respectively. The poly-mineral coarse-grained material was mounted on stainless

steel disc using silicon oil. The aliquot size was kept to 5 mm which is equivalent to a few hundred
grains per aliquot. Altogether, three aliquots per sample were measured. More details of the
measurement protocol, the experimental set-up and rejection criteria are provided in Reimann et al.
(2015). To calculate the reduction of dose we used the threshold for the onset of dose saturation
2D0 (Wintle and Murray, 2006), which is equivalent to 85 % full signal saturation, for

normalization. The normalized reduction for each sample to this threshold was calculated from
three aliquots and based on the un-weighted mean and the 1-sigma standard error as uncertainty.

To test the performance of the PMS approach especially with regard to blue stimulated OSL at
125°C (BSL-125), which is supposed to be dominated by quartz OSL fast-component, we density-
separated 16 out of 38 samples using LST and applied etching with 40 % HF for 45 min to

chemically removed any contamination of feldspar luminescence to the BSL-125 signal. The BSL-
125 signal of the purified and etched quartz extracts was measured applying a conventional quartz
OSL SAR protocol (Wintle and Murray, 2006) to typically 2 mm aliquots, equivalent to ~50 grains





(Table 3). The normalized reduction compared to 2D0 was calculated for each sample from three aliquots and based on the unweighted mean. The 1-sigma standard error as uncertainty was chosen.

## 3.    Results

### 3.1.    Subaqueous light spectra

Our measurements show a marked dependency of light attenuation on the suspended sediment concentration that varies over the wavelength interval of 315–930 nm (Fig. 3a). Clear water slightly reduces the intensity in the UV (<400 nm) but has virtually no impact within the 400–500 nm range. From 500 to 650 nm the effect is of similar magnitude as for the UV, but particularly in the wavelength range of 650–800 nm the attenuation is twice as high. Upon addition of just 10 g l$^{-1}$ of sediment the irradiance in the UV/blue region (315 – 450 nm) drops sharply by 77 %, whereas the drop in the red/NIR region (625 – 900 nm) is 48 % and thus similar to that caused by clear water alone. With increasing sediment concentration this trend continues until the UV/blue component is effectively canceled out. Moreover, the spectral peak successively broadens and shifts from ~500 nm under subaerial conditions to ~600 nm with a $C_s$ of 40 g l$^{-1}$. When increasing the sediment concentration even further, up to 100 g l$^{-1}$ (Fig. 3b), the wavelengths smaller than 500 nm almost disappear and the spectral peak moves to ~710 nm. So, in this experiment, we observe that (1) clear water attenuates the IR part of the spectrum stronger than UV/blue, (2) the reduction of light intensity with increasing $C_s$ is wavelength-dependent, (3) UV/blue is fully attenuated with $C_s$ greater than 40 g l$^{-1}$ and (4) the spectral peak shifts to the red/NIR region. The reduction of light intensity follows an exponential decay of the form:

$$I = I_0 \cdot e^{-\lambda \cdot Cs}, \tag{1}$$

where the decay constant λ is a linear function of the wavelength (Fig. 3c).

This $C_s$-dependent modification of the light spectrum is also reflected within the calibrated irradiance-time series that we measured simultaneously with the bleaching of the samples (Fig. 4). We further observe a gradual decline in the intensity reading of the instrument with time that is superimposed by sudden intensity drops. The former can be attributed to the lowering of the sun with the progression of the afternoon and the latter to shadows from clouds. Consequently, when plotting the total received irradiance against the bleaching time, we observe a decrease of the relative contribution of the last bleaching interval, in particular in the 33 g l$^{-1}$ case. Thus, it may be more appropriate to relate the measured luminescence dose to the total received irradiance rather than to the bleaching duration. Comparing the irradiance time-series we note that with 66 g l$^{-1}$ we measure a lower irradiance than with 100 g l$^{-1}$. This is due to more overcast conditions on this day, which is also registered in the subaerial reference spectra that we measured before and after each





experiment (Fig. 5). At the end of the last illumination interval with $C_s$ = 66 g l$^{-1}$, the magnet of the stirrer stopped spinning which led to rapid settling of the grains and a sudden drop in turbidity. Therefore, we omitted the last sample of the 66 g l$^{-1}$ run from further analyses.

In the following we investigate, whether the above changes in the light characteristics had an effect
on the differential bleaching of quartz and feldspar during the experiments.

### 3.2. Bleaching behavior

We initially limited our analysis to the polymineral multiple signal protocol (see methods) as a time-effective means of profiling the luminescence signal loss with bleaching. In all samples of the subaqueous experiments, the PMS BSL-125 bleaches slower than the IR-25 and the pIRIR-155
(Fig. 6), which is consistent with the theoretical considerations outlined in the introduction and with the evidence presented in section 3.1 and Fig. 3. However, in our subaerial control run we observed the same pattern of slow BSL-125 using the PMS approach, which contradicts our theoretical considerations and the previous experimental work (e.g. Godfrey-Smith et al. 1988; Buylaert et al. 2012). To investigate, whether this unexpected observation is caused by contaminated BSL-125
signals within the PMS approach, we additionally measured the BSL-125 on purified quartz for a subset of the samples. In contrast to the BSL-125 within the PMS protocol, the genuine quartz yields a BSL-125 that decays faster than the feldspar signals in all experiments. A possible explanation for this observation is elaborated on in the discussion section below.

Among all our experiments, the subaerial run features the lowest overall luminescence response
with respect to 2D0 (Fig. 6a). After six seconds of illumination the normalized BSL-125 dose is already bleached to 2 %. After five minutes it further decreases by about one order of magnitude but remains near constant thereafter. We detect a similar pattern for the IRSL-25 and the pIRIR-155 signals, i.e. large decreases of the normalized dose within the initial five minutes and only minor changes afterwards. In general, with increasing $C_s$ more of the normalized dose remains for all the
three signals (Fig. 7a-c). Thus, the turbidity modulates the energy flux density received by the grains as has been expected. The disproportionally larger discrimination against short wavelengths by the turbid suspension (Fig. 3), however, does not cause an inversion of the relative responses of the blue-stimulated pure quartz and the IR-stimulated luminescence signals. This is at odds with our hypothesis that there exists a turbidity condition, at which both signals decay at the same rate.

When relating the luminescence response to the cumulative irradiance instead of the bleaching time, a very different pattern appears (Fig. 7d-f). A general trend of decreasing luminescence response with increasing cumulative irradiance is clearly captured. From a theoretical perspective though, samples that have received the same amount of irradiance should bear the same remaining luminescence given everything else being equal. But in our case, differences in the remaining





luminescence of up to one order of magnitude were measured for samples, which received a highly similar cumulative irradiance. Given the relatively low error on the normalized dose this scatter cannot entirely be attributed to uncertainties of the luminescence measurements. It may thus follow from methodological limitations of the chosen experimental setup. A possible explanation is that some transient settling of grains occurred at the bottom of the container and that the spectrometer
measurements therefore registered a too high cumulative irradiance. With increasing sediment concentration this effect presumably increased as well, which would explain the systematically higher residual doses measured for the 66 g l⁻¹ and 100 g l⁻¹ samples with respect to the 33 g l⁻¹ samples.

## 4. Discussion

### 4.1. Why so fast?

The measurements of the subaqueous irradiance demonstrate that the addition of sediment has a profound effect on the intensity and wavelength distribution of light within the water column. With increasing sediment concentration, the blue portion of the spectrum diminishes and eventually fades into measurement noise, whereas the intensity peak shifts from ~500 nm to ~700 nm, i.e. to the
red/NIR region. The wavelength shift has been observed in previous experiments as well (e.g. Berger and Luternauer, 1987; Sanderson et al. 2007). Bleaching of quartz is predominantly caused by the blue wavelength (~470 nm) spectrum (Spooner 1994b), which raises the question, why the BSL-125 still decays relatively fast with respect to the IR-25 and pIRIR-155 (Fig. 6). A possible explanation relates to our experimental setup, where the light characteristics are continuously
recorded at a fixed position inside the flume. Yet, the turbulent flow inside the container causes mixing of grains throughout the water column. Thus, there is a high probability that a large fraction of grains will sporadically appear at the water surface during the experiment. Albeit short, these exposures to an unaltered solar spectrum may well be sufficient to bleach the quartz (Gray et al. 2017). Although this is true for the feldspar as well, the effect will be more pronounced for the
quartz signal as even a quick exposure to UV/blue light can more efficiently reset the BSL-125 signal (Godfrey-Smith et al., 1988). To test, whether these bleaching patterns also dominate in natural settings, modifications of our experimental setup need to be incorporated to ensure more laminar flow and less vertical mixing, which would provide the means to investigate the effect of turbulence. Yet such modifications were beyond the scope of this study. However, our data clearly
hints at the importance of flow turbulence as a potential control of luminescence signal bleaching.

### 4.2. Turbidity versus suspended sediment concentration

It is important to note that $C_s$ is only a proxy for the turbidity (e.g. Minella et al., 2008) and that the relation of both is a function of the sample composition, mineralogy and grain-size distribution





(Davies-Colley and Smith, 2001; Merten et al., 2014). For example, when we decant our sample
and refill the container with clear tab water the turbidity is gone although the mass loss of the
sample is minor. Thus, only a tiny fraction of the sediment sample determines the turbidity of the
water. This fraction is predominantly composed of silty and clayey particles of yet unknown
mineralogy and probably contains some organic material. This implies that a repeat bleaching
experiment with a different sample would probably result in different luminescence responses (at
least for the feldspar components) even if sediment concentration and boundary conditions were
held constant.

### 4.3. Light spectra measurements

As shown in section 3.1 and Fig. 3, the decrease in irradiance follows an exponential decay that is
controlled by the sediment concentration and a wavelength-dependent decay constant, λ. Thus, the
knowledge of λ enables the computation of the energy flux density for a broad range of wavelengths
and sediment concentrations, which may be useful for the modelling of luminescence signal
resetting in turbid suspensions. Because the turbidity is controlled not only by the sediment
concentration but also by other sediment characteristics (see section above), we infer that λ is as
well sample-specific. However, in natural systems water depth, salinity, stratification and/or vertical
turbidity gradients may also affect the energy flux density but are not captured in Eq. 1. Thus, the
developments of flume experiments that enable isolating these individual factors are an avenue for
further research.

### 4.4. Performance of the PMS approach for capturing bleaching trends

Previous studies suggest that the PMS approach is able to sufficiently capture dose trends also with
regard to the BSL-125 signal (e.g. Reimann et al. 2015; Chamberlain et al. 2017). The advantage of
the approach is a very time efficient measurement of larger sets of samples and in particular the
prospect of measuring multiple luminescence signals – with various bleaching rates – from the same
sub-sample of mineral grains. For our samples, however, the approach was not proficient to
measure a clean-enough quartz dominated BSL-125 signal from minimally prepared samples. A
comparison to a sub-set (16 samples) of fully purified quartz extracts may suggest that the PMS
based BSL-125 signal is likely contaminated by feldspar signals components that were not removed
during the IR (at 25 °C) and pIRIR wash (at 155 °C, and 225 °C). This observation is surprising as
previous studies show that multiple IR wash including multiple steps at elevated temperature are
able to remove all feldspar luminescence signals from a feldspar contaminated sample (e.g. Zhang
and Zou, 2007). An alternative explanation could be that the minimally prepared samples are
contaminated by other luminescent minerals (e.g. heavy minerals) that are not sufficiently sensitive
to the IR and/or elevated temperature pIRIR stimulation. To test the latter hypothesis one could





further purify a subset of minimally prepared sample by removing the heavy mineral for example through density and/or magnetic separation (Porat, 2006) and compare the corresponding BSL-125

results with those of the minimally prepared PMS samples and the fully prepared pure quartz extracts. For future studies using the PMS approach it seems advisable to test the performance of the corresponding BSL-125 signal by using a BSL-125 measurement on a subset of fully prepared quartz extracts as benchmark.

### 4.5.   Implications for using luminescence as sediment transport tracer

The use of luminescence as sediment transport tracer is so far mainly based on the sensitivity of sand grain luminescence signal bleaching to sediment transport lengths and the mean resting time of the grains between two transport episodes (e.g. Gray et al. 2017, 2019; Guyez et al. 2023). For example, Guyez et al. (2023) were able to compute the virtual transit velocity of sand grains passing through two braided river systems in New Zealand from single-grain feldspar pIRIR data. The aim

of our experiments was to systematically investigate the sensitivity of bleaching of different luminescence signals (quartz OSL vs. feldspar IR/pIRIR) to changes in turbidity to evaluate the potential of luminescence sediment transport tracing to determine palaeoturbidities from fluvial sediment records such as river terrace sequences. Interestingly, however, our data is not showing the threshold $C_s$ we hypothesize, beyond which feldspar IR/pIRIR signals bleach faster than quartz

OSL.

The interpretation of this observation is not straightforward. Our light spectra experiment clearly show the impact of turbidity on the effective intensity but also show a shift of the spectrum towards longer wavelengths, which potentially favour the bleaching of feldspar IR/pIRIR signals. Yet, the reduction of the light intensity (for $C_s = 10$ g l$^{-1}$ only 30 % of light intensity is left) is far more

significant than the changes in the relative contribution of longer wavelengths. This supports the finding that the bleaching efficiency for sediments in a turbid water column drops dramatically even for relatively moderate sediment concentration (Ditlefsen, 1992). Hence, the effective bleaching of the grains is likely controlled by the short periods the grains spend at the surface and receive nearly subaerial bleaching conditions. For our experimental set-up bleaching of the grains is thus mainly

controlled by the vertical mixing velocity (i.e. turbulence), which explains why we did not observe a convergence of quartz OSL vs. feldspar IR/pIRIR bleaching rates.

## 5.   Conclusions

The aim of this study was to devise and conduct an experiment that helps to evaluate the effect of turbidity on the bleaching of fluvially transported sediment. We found a systematic relationship

between the suspended sediment concentration and the irradiance at wavelength between 300 – 900



nm. Increases in $C_s$ lead to a stronger attenuation of the UV/blue compared to the red/NIR wavelength i.e. the shorter the wavelength, the faster it disappears within the suspension and vice versa (Fig. 3). The broadband, $C_s$-dependent decrease in irradiance was captured in the luminescence response with the remaining normalized doses being positively correlated with $C_s$

(Fig. 7a-c). The sediment concentration thus modifies the resetting time for the measured BSL-125, IRSL-25 and pIRIR-155 signals. However, we did not find the wavelength-dependent discrimination of light in the suspension to be reflected in the luminescence data. Particularly, the hypothesized threshold in $C_s$, at which the BSL-125 and the IRSL-25 signals bleach at the same rate, could not be established. We attribute this to the turbulence and vertical mobility of grains in

the flume, which may lead to a fast bleaching of BSL-125 during short and sporadic moments when grains emerge at the water surface. This implies that turbulent flow conditions are an important prerequisite for bleaching grains during transport. Yet, another experiment would have to be designed to further scrutinize this finding.

The PMS approach resulted in a slower bleaching of the BSL-125 when compared to the IRSL-25,

even under subaerial conditions (Fig. 6). The analysis on purified quartz showed the opposite behaviour of BSL-125, which rendered the PMS based BSL-125 unreliable in this case. A potential reason could be the presence of other luminescent minerals (e.g. heavy minerals) that contaminated the BSL-125 signal. Thus, for future studies using the PMS, we recommend to benchmark the BSL-125 using a subset of fully purified quartz extracts.

Particularly, in bleaching experiments that use daylight conditions and long illumination times, irradiance will be subject to gradual as well as abrupt variations. Therefore, it seems advisable to relate the measured luminescence dose to the cumulative received irradiance rather than to the bleaching time. For a more complete picture of the irradiance boundary conditions it may even be better to employ a second spectrometer that simultaneously registers the incoming subaerial

radiation.

*Code and data availability.* The code and data to run the analysis are available in the supplemental data.

*Author contribution.* JM, WS and TR conceived the study, JM conducted the experiment and AMdB performed the sample preparation and luminescence measurements. All authors wrote the

manuscript.

*Competing interests.* The authors declare that they have no conflict of interest.



*Acknowledgements.*

This project was financed by the "Experiment!" funding initiative of the VolkswagenStiftung

(Project: Illuminating the speed of sand). We thank W. Stackebrandt for suggestions on the

sampling site, and S. Tofelde for help during the field excursion. J.M. expresses his gratitude to J.

and R. Heinrich for providing the outdoor environment during the Covid-19 pandemic.

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





## Figures

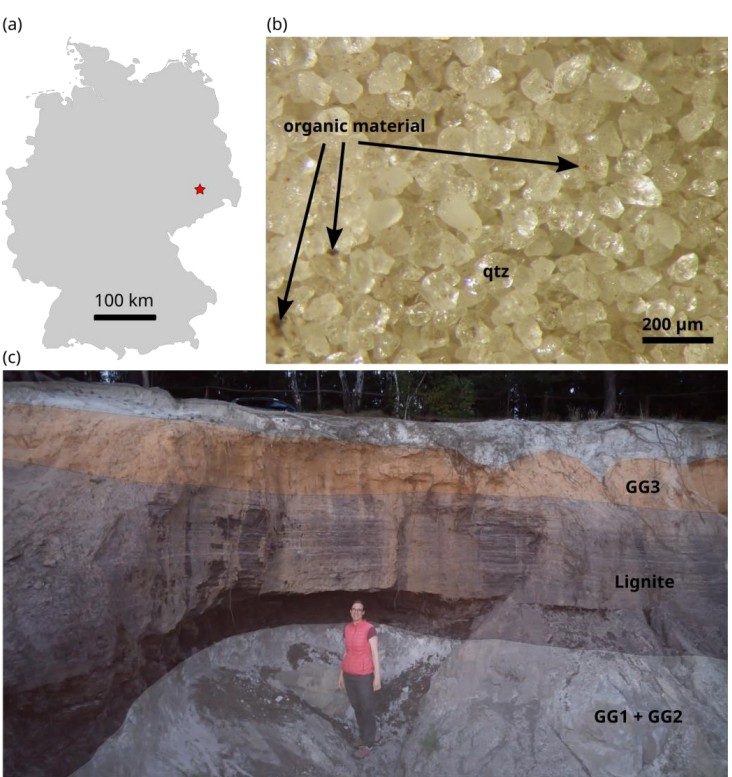

**Fig. 1. Sample location.** We sampled coastal sand of Miocene age (Stackebrandt and Franke, 2015) from an abandoned open coal pit in southern Brandenburg (a). Two sand layers, above and below a lignite horizon, where sampled (c). All experiments were conducted using material from sample GG1, which is of homogeneous and near-monomineralic character (b). Dark brown material is attached to some grains, which presumably plays an important role in creating turbidity. The person in (c) is 170 cm tall.





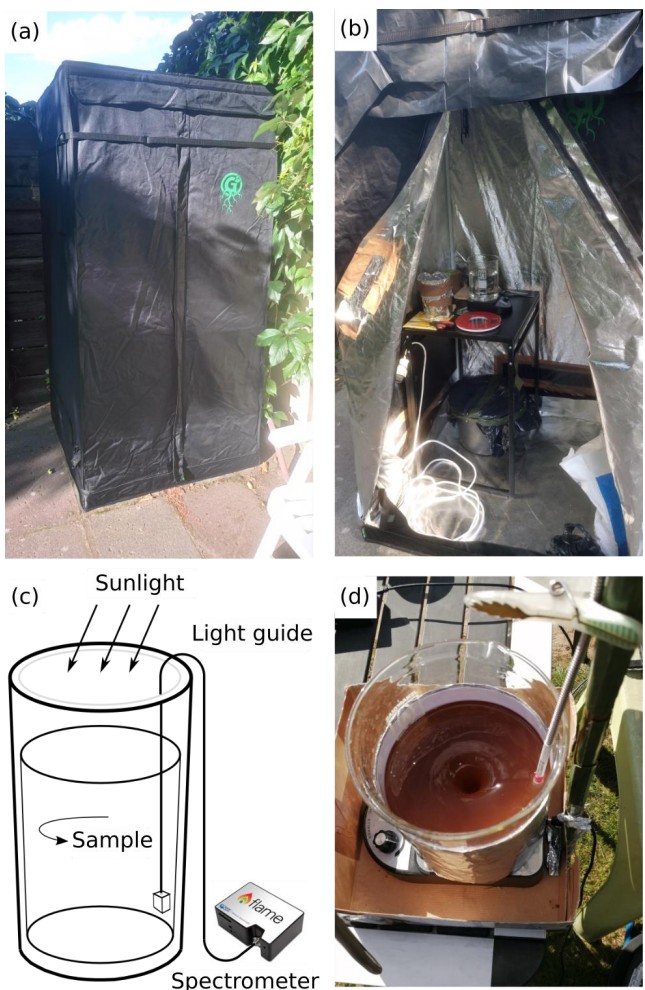

**Fig. 2. Experimental setup.** We used a circular flume (c) and a light-proof tent (a) in an outdoor laboratory (b). Subaqueous solar irradiance was measured using a UV–IR spectrometer with a submersible probe (d). We conducted experiments with suspended sediment concentrations of 10, 33, 66, and 100 g l$^{-1}$ and illuminated for time intervals of 5 minutes to 5 hours, respectively. After each exposure interval a subsample was extracted for OSL/IRSL analysis in the light-proof tent.






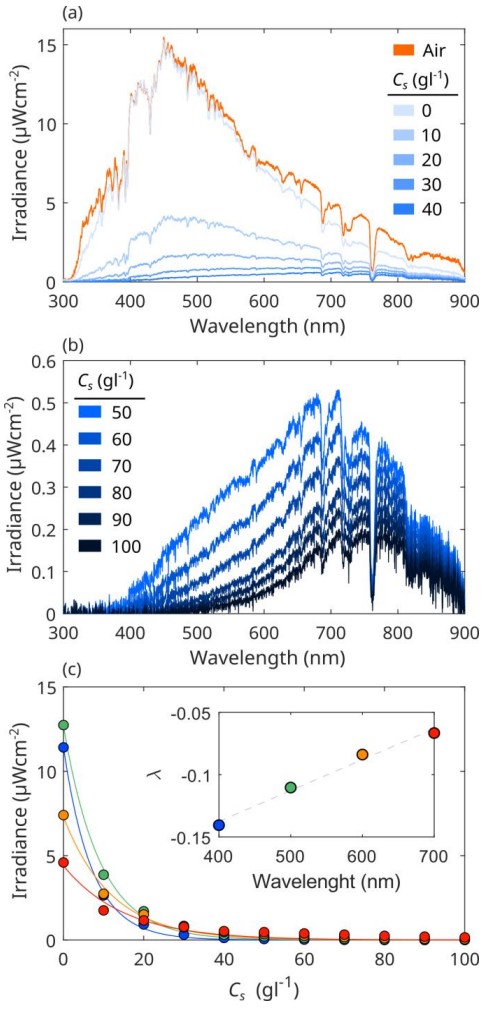

**Fig. 3. Spectra in turbid suspensions.** (a) Subaerial and clear water (0 $gl^{-1}$) spectra as a reference for the measurements with increased suspended sediment concentrations ($C_s$) of 10–40 $gl^{-1}$ and (b) 50–100 $gl^{-1}$. (c) The attenuation of light follows an exponential decay with the decay constant λ being a function of the wavelength. In high turbidity conditions shorter wavelengths disappear more quickly and the spectral peak broadens and shifts to the red/NIR region.



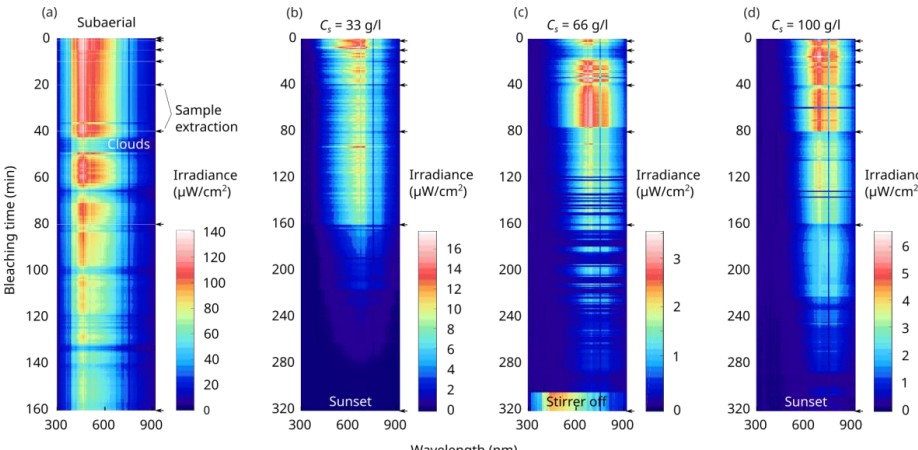

**Fig. 4. Irradiance time-series.** We continuously measured the in situ irradiance received by the grains in the wavelength interval of 300–915 nm for (a) the subaerial case and with a $C_s$ of (b) 33 g l$^{-1}$, (c) 66 g l$^{-1}$ and (d) 100 g l$^{-1}$. With increasing $C_s$ the irradiance decreases and the spectral peak broadens and shifts to the longer wavelength. Note the different bleaching-time scales of the subaerial and the subaqueous experiments. At the end of the last bleaching interval in (c) the magnet of the stirrer lost connection and turbidity decreased due to the settling of grains. $C_s$: suspended sediment concentration.





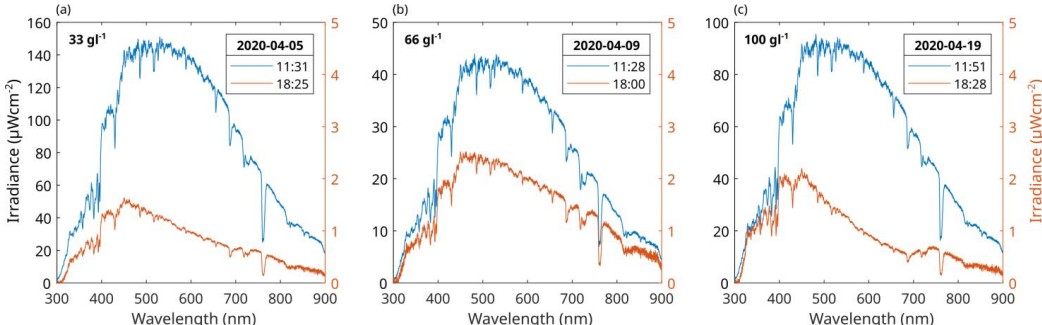

**Fig. 5. Subaerial spectra.** Before and after each experiment, we recorded the subaerial spectra for reference. Note that in the 66 gl$^{-1}$ experiment (b) the noon spectrum features an irradiance peak smaller than 50 µWcm$^{-2}$. This is due to a generally more overcast sky from noon to early afternoon on that day.

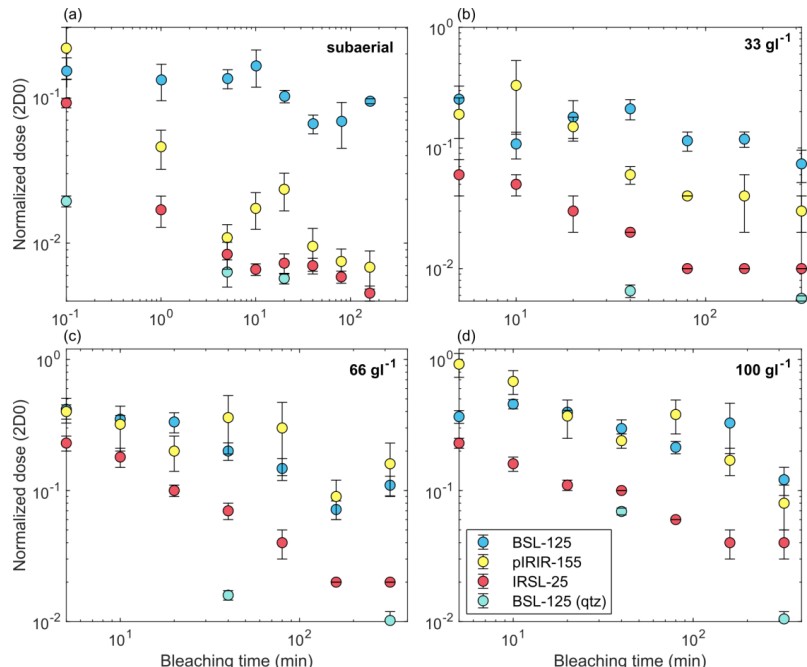

**Fig. 6. Bleaching behaviour I.** Polymineral luminescence response of BSL-125, pIRIR-155 and IRSL-25 as well as BSL-125 of pure quartz as a function of the daylight bleaching time for all experiments. Note the slow bleaching of the polymineral BSL-125 in the subaerial case.






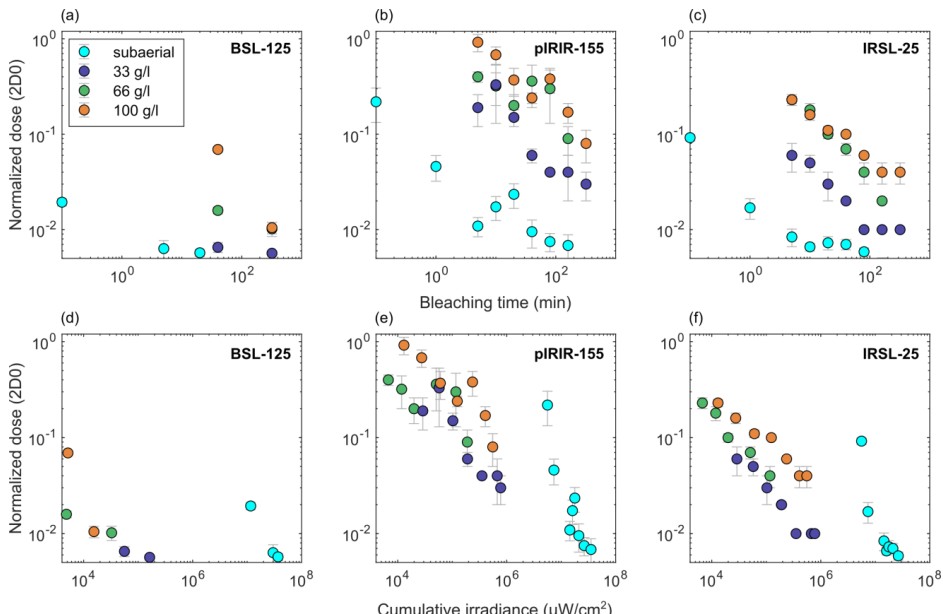

**Fig. 7. Bleaching behaviour II.** Bleaching time (a-c) and cumulative irradiance (d-f) versus the normalized dose shown for the pure quartz BSL-125 and the feldspar-dominated pIRIR-155 and IRSL-25. Particularly panels (b) and (c) show that increased $C_s$ is systematically accompanied with larger remaining normalized doses, i.e. a slower signal resetting. For the BSL and IR in panels (d) – (f), we integrated over the wavelength intervals of 300 – 600 nm and 600 – 900 nm, respectively.





# Tables

**Table 1:** Experiments conducted in this study.

| $C_s$ (g/l)[1] | Total illumination time (min) | Date | Time of day |
|---|---|---|---|
| 33 | 5, 10, 20, 40, 80, 160, 320 | 2020-04-05 | 12:00-18:00 |
| 66 | 5, 10, 20, 40, 80, 160, 320 | 2020-04-09 | 12:00-18:00 |
| 100 | 5, 10, 20, 40, 80, 160, 320 | 2020-04-19 | 12:00-18:00 |
| subaerial | 0.1, 1, 5, 10, 20, 40, 80, 160 | 2021-05-09 | 13:00-17:00 |

[1]suspended sediment concentration


**Table 2:** PMS luminescence measurements.

| Lab code | Cs (g l$^{-1}$) | Bleaching time (min) | De [Gy] | | | Norm. 2D0 | | |
|---|---|---|---|---|---|---|---|---|
| | | | BSL-125 | IRSL-25 | pIRIR-155 | BSL-125 | IRSL-25 | pIRIR-155 |
| NCL-1220078 | 33 | 5 | 71.3 ± 20.3 n = 3 | 40.2 ± 10.3 n = 5 | 46.0 ± 16.6 n = 3 | 0.25 ± 0.07 | 0.06 ± 0.02 | 0.19 ± 0.07 |
| NCL-1220079 | 33 | 10 | 30.4 ± 7.6 n = 4 | 29.9 ± 5.4 n = 6 | 79.5 ± 47.0 n = 4 | 0.11 ± 0.03 | 0.05 ± 0.01 | 0.33 ± 0.20 |
| NCL-1220080 | 33 | 20 | 50.7 ± 18.7 n = 6 | 21.8 ± 4.0 n = 6 | 35.1 ± 6.1 n = 6 | 0.18 ± 0.07 | 0.03 ± 0.01 | 0.15 ± 0.03 |
| NCL-1220081 | 33 | 40 | 59.4 ± 11.3 n = 4 | 12.9 ± 1.7 n = 6 | 14.5 ± 2.9 n = 4 | 0.21 ± 0.04 | 0.02 ± 0 | 0.06 ± 0.01 |
| NCL-1220082 | 33 | 80 | 32.3 ± 5.8 n = 5 | 9.6 ± 0.3 n = 6 | 8.59 ± 0.70 n = 5 | 0.11 ± 0.02 | 0.01 ± 0 | 0.04 ± 0 |
| NCL-1220083 | 33 | 160 | 33.3 ± 4.8 n = 6 | 7.2 ± 0.7 n = 6 | 9.07 ± 4.10 n = 6 | 0.12 ± 0.02 | 0.01 ± 0 | 0.04 ± 0.02 |
| NCL-1220084 | 33 | 320 | 20.8 ± 6.2 n = 6 | 6.0 ± 0.8 n = 6 | 8.20 ± 2.00 n = 6 | 0.07 ± 0.02 | 0.01 ± 0 | 0.03 ± 0.01 |
| NCL-1220085 | 66 | 5 | 117 ± 25 n = 6 | 146 ± 19 n = 6 | 95.5 ± 12.3 n = 6 | 0.42 ± 0.09 | 0.23 ± 0.03 | 0.4 ± 0.05 |
| NCL-1220086 | 66 | 10 | 98.5 ± 6.7 n = 5 | 114 ± 22 n = 6 | 77.9 ± 29.7 n = 5 | 0.35 ± 0.02 | 0.18 ± 0.03 | 0.32 ± 0.12 |
| NCL-1220087 | 66 | 20 | 93.8 ± 16.4 n = 6 | 62.0 ± 5.9 n = 5 | 47.9 ± 14.2 n = 6 | 0.33 ± 0.06 | 0.10 ± 0.01 | 0.20 ± 0.06 |
| NCL-1220088 | 66 | 40 | 56.3 ± 8.6 n = 5 | 43.15 ± 6.2 n = 6 | 86.9 ± 39.7 n = 5 | 0.20 ± 0.03 | 0.07 ± 0.01 | 0.36 ± 0.17 |
| NCL-1220089 | 66 | 80 | 41.4 ± 7.9 n = 5 | 26.8 ± 3.4 n = 5 | 73.0 ± 41.2 n = 5 | 0.15 ± 0.03 | 0.04 ± 0.01 | 0.30 ± 0.17 |
| NCL-1220090 | 66 | 160 | 20.2 ± 3.3 n = 6 | 13.6 ± 2.0 n = 6 | 22.3 ± 6.3 n = 6 | 0.07 ± 0.01 | 0.02 ± 0 | 0.09 ± 0.03 |
| NCL-1220091 | 66 | 320 | 30.9 ± 5.3 n = 5 | 15.7 ± 2.7 n = 6 | 39.4 ± 17.3 n = 5 | 0.11 ± 0.02 | 0.02 ± 0 | 0.16 ± 0.07 |



| NCL-1220092 | 100 | 5 | 103 ± 12  n = 6 | 146 ± 16  n = 6 | 220 ± 45  n = 6 | 0.37 ± 0.04 | 0.23 ± 0.02 | 0.92 ± 0.19 |
|---|---|---|---|---|---|---|---|---|
| NCL-1220093 | 100 | 10 | 128.4 ± 10.4 n = 6 | 106 ± 15  n = 5 | 162 ± 32  n = 6 | 0.46 ± 0.04 | 0.16 ± 0.02 | 0.68 ± 0.14 |
| NCL-1220094 | 100 | 20 | 111.4 ± 3.9 n = 5 | 68.4 ± 4.2 n = 6 | 88.6 ± 29.0 n = 5 | 0.40 ± 0.01 | 0.11 ± 0.01 | 0.37 ± 0.12 |
| NCL-1220095 | 100 | 40 | 83.5 ± 13.8 n = 6 | 64.1 ± 2.1 n = 5 | 58.2 ± 7.4 n = 6 | 0.30 ± 0.05 | 0.1 ± 0.0 | 0.24 ± 0.03 |
| NCL-1220096 | 100 | 80 | 60.1 ± 6.5 n = 5 | 39.5 ± 2.1 n = 5 | 91.3 ± 25.4 n = 5 | 0.21 ± 0.02 | 0.06 ± 0.0? | 0.38 ± 0.11 |
| NCL-1220097 | 100 | 160 | 92.1 ± 38.3 n = 5 | 25.00 ± 4.0 n = 6 | 40.6 ± 10.8 n = 5 | 0.33 ± 0.13 | 0.04 ± 0.01 | 0.17 ± 0.04 |
| NCL-1220098 | 100 | 320 | 34.0 ± 8.3 n = 6 | 23.9 ± 5.5 n = 3 | 19.4 ± 7.3 n = 6 | 0.12 ± 0.03 | 0.04 ± 0.01 | 0.08 ± 0.03 |
| NCL-1621077 | subaerial | 0.1 | 34.6 ± 9.1 n = 6 | 54.9 ± 5.00 n = 6 | 43.0 ± 11.1 n = 5 | 0.15 ± 0.03 | 0.092 ± 0.007 | 0.22 ± 0.09 |
| NCL-1621078 | subaerial | 1 | 30.7 ± 13.0 n = 5 | 12.9 ± 2.9 n = 6 | 10.8 ± 3.6 n = 5 | 0.13 ± 0.04 | 0.017 ± 0.004 | 0.05 ± 0.01 |
| NCL-1621079 | subaerial | 5 | 38.7 ± 5.5 n = 6 | 8.15 ± 1.12 n = 6 | 2.27 ± 0.46 n = 5 | 0.13 ± 0.02 | 0.008 ± 0.002 | 0.011 ± 0.003 |
| NCL-1621080 | subaerial | 10 | 41.4 ± 7.9 n = 5 | 5.80 ± 0.70 n = 6 | 5.06 ± 1.77 n = 6 | 0.16 ± 0.05 | 0.007 ± 0.001 | 0.017 ± 0.005 |
| NCL-1621081 | subaerial | 20 | 37.8 ± 5.6 n = 6 | 4.59 ± 0.81 n = 5 | 5.26 ± 1.97 n = 6 | 0.10 ± 0.01 | 0.007 ± 0.001 | 0.024 ± 0.007 |
| NCL-1621082 | subaerial | 40 | 15.7 ± 3.0 n = 5 | 4.34 ± 0.59 n = 6 | 1.45 ± 0.21 n = 6 | 0.07 ± 0.01 | 0.007 ± 0.001 | 0.010 ± 0.003 |
| NCL-1621083 | subaerial | 80 | 15.6 ± 6.4 n = 4 | 5.89 ± 2.82 n = 5 | 2.43 ± 1.11 n = 4 | 0.07 ± 0.02 | 0.006 ± 0.001 | 0.008 ± 0.002 |
| NCL-1621084 | subaerial | 160 | 20.1 ± 0.8 n = 3 | 2.84 ± 0.35 n = 6 | 1.54 ± 0.55 n = 6 | 0.09 ± 0 | 0.005 ± 0.001 | 0.007 ± 0.002 |

**Table 3:** Genuine quartz luminescence measurements (BSL-125).

| Lab code | Cs (g l⁻¹) | Bleaching time (min) | De [Gy] | Norm. 2D0 |
|---|---|---|---|---|
| NCL-1220081 | 33 | 40 | 0.99 ± 0.14 n = 6 | 0.0065 ± 0.000777 |
| NCL-1220084 | 33 | 320 | 0.92 ± 0.10 n = 6 | 0.0057 ± 0.000286 |
| NCL-1220088 | 66 | 40 | 3.33 ± 0.58 n = 6 | 0.0159 ± 0.0013 |
| NCL-1220091 | 66 | 320 | 1.64 ± 0.27 n = 6 | 0.0102 ± 0.0017 |
| NCL-1220095 | 100 | 40 | 10.92 ± 1.13 n = 6 | 0.0693 ± 0.0041 |
| NCL-1220098 | 100 | 320 | 1.59 ± 0.25 n = 6 | 0.0105 ± 0.0014 |
| NCL-1621077 | subaerial | 0.1 | 22.92 ± 2.20 n = 16 | 0.0194 ± 0.0017 |



| NCL-1621079 | subaerial | 5 | 6.86 ± 1.53 n = 16 | 0.0063 ± 0.0013 |
|---|---|---|---|---|
| NCL-1621081 | subaerial | 20 | 7.89 ± 0.82 n = 16 | 0.0057 ± 0.00483 |