# Peer review of "Differential bleaching of quartz and feldspar luminescence signals under high turbidity conditions"

_Geochronology, 2023_

## Author Response (AR1)

We thank the reviewer H. Gray for his constructive comments. Below, we provide replies to the main comment and some of the minor comments and detail how we will address them in a revised version of the manuscript.

—----------------
R1:

In *Differential bleaching of quartz and feldspar luminescence signals under high turbidity conditions* by Mey et al., the authors describe an experiment to determine the effect of variable subaqueous suspended sediment concentration on sunlight irradiance and the bleaching (resetting) rates of quartz and feldspar luminescence. The authors constructed an experimental set up consisting of a chamber to control sunlight exposure surrounding a filled beaker with magnetic stir bar and submerged light spectrometer. The authors then performed experiments in which they measured the resulting irradiance and luminescence bleaching as a function of sediment concentration. They discovered that increasing suspended sediment concentration leads to larger residual doses and that differential attenuation of light by wavelength did not appear to have a notable effect. The authors then concluded that turbulent upwelling leading to grains reaching the water's surface probably resulted in most of the overall bleaching.

Overall, I think this work is a useful contribution. The effect of differential attenuation of light by wavelength is a notable step towards answering the broader problem of where and how bleaching occurs in natural fluvial environments. The latter question being an important topic of study within the development of luminescence as a sediment tracer. I also like the author's point that cumulative irradiance is more important towards bleaching than overall exposure time.

I have minor comments about the experimental set up, but nothing so large as to require performing the experiment over. I agree with the basic mechanics of this experiment (to be fair, I did a very similar experiment during a past study (Gray et al. 2018, Supplemental material: Figure S7)). I would have liked the authors to have gone further with this data, perhaps coupling it with a numerical model simulating particle trajectories in fluvial turbulence or similar, but time is limited and this contribution is worth entering into the public domain.

**Main Comment**

In the experiment, the authors use suspended sediment concentrations of 33, 66, and 100 grams per liter. These seem like very very high sediment concentrations. My experience has been that suspended sediment concentrations in rivers are typically 1-2 orders of magnitude lower than used in this experiment (for example John Gray and Francisco Simões 2008's manual). Certainly, the authors have identified the concentrations of fairly extreme flood events, but I think the authors should state that these may not be reflective of the general conditions of sediment transport in rivers.

For example, in some river systems, the majority of the sediment is transported in Wolman and Miller (1960) style 'effective discharge' conditions, which are a balance between the frequency and magnitude of flooding events. Very large floods with high sediment concentrations, such as those simulated here, move a large amount of sediment, but their

effects are offset by their infrequency. In contrast, medium-sized floods that optimize the frequency and magnitude of sediment transport ('effective discharge events') end up doing the most work in the system. I think it is worth considering that these effective discharge events represent the conditions which need to be simulated in order for these experiments to have the most relevance for luminescence sediment tracing.

A:
We agree that the used sediment concentrations may be at the upper end of concentrations that have been measured in most rivers. However, in active orogens like the Himalayas with high rainfall intensity, landslide activity and a strong coupling between hillslopes and channels, such concentrations may be reached more often. For example Wulf et al. (2012) show that in the winter of 2007, SSC in the Baspa river, a tributary to the Sutlej river, equaled or exceeded 80 g/l twice and the Sutlej itself reached 60 g/l (https://doi.org/10.5194/hess-16-2193-2012, their Fig. 6B). Thus we can assume that the "effective discharge", at least in such regions, is indeed associated with extremely high SSCs. Nevertheless, we will follow the suggestion of the reviewer and specify more clearly what kind of environment we are simulating.
Just a note from the practical perspective: Working with SSCs of less than 10 g/l would increase the relative bias that is introduced by removing parts of the sample after each illumination interval. Although doable, the entire workflow for such an experiment would have to be redesigned.

We will address the above points in more detail in a revised version of the manuscript and incorporate the suggested minor edits.

**Minor edits:**

It would be helpful to have a Table of the PMS protocol

We added this table to the supplementary material.

Line 41: "anisotropies" doesn't seem exactly right. How about "heterogeneities" ?

changed

Line 60: Stokes et al., 2001 (quartz) and Gray et al, 2018 (feldspar) have downstream bleaching trends as well

added

Line 111: "tab water" -> "tap water"

corrected

Line 109: This experimental setup is similar to that of Gray et al. 2018 (see, supplemental material). It may be worth mentioning that study to provide support for the conception of the experimental apparatus shown here.

(A) Agreed.

Line 124: Does the sunlight beam setup (non reflecting sides) result in an overall lower radiance? I wonder because light scattering in natural environments might change the flux from various wavelengths if different wavelengths are scattered differently.

(A) Yes, excluding light penetration from the sides does lower the overall radiance. We wrapped the container to avoid contamination of the light by scattering from anthropogenic surfaces, e.g. buildings. But more importantly, light from the sides would directly hit the grains that are traveling along the inner wall of the container, which may offset the imposed turbidity.

Line 127: See main comment above.

Line 270: "tab water" -> "tap water"

corrected

Line 297-300: I have also had this problem when I tried the PMS approach on a project a few years ago.

Also general observations with PMS:

One thing to consider is the possibility that the heating during the IR steps are bleaching/modifying the BSL signal prior to measurement. Potentially moving the fast component electrons into slower components. This happens to us here in the western USA because the quartz is often thermally unstable. ALSO with the polymineral approach, the relative qtz/feldspar ratio may matter as the IR-insensitive quartz grains may somewhat shield the feldspar from the radiation source in the Risoe. I had this happen on a quartz/feldspar mixture and purifying the feldspar fixed the problem.

(A) Many thanks for sharing these valuable thoughts with us. Regarding qtz/feldspar ratio: our data - especially the comparison of fully prepared qtz OSL measurements and polymineral PMS measurements - suggests that the problem is most likely related to the qtz data and not the feldspar.

Figure 3c: "wavelenght" -> "wavelength"

corrected

**References**

Gray, H. J., Tucker, G. E., & Mahan, S. A. (2018). Application of a luminescence‑based sediment transport model. Geophysical Research Letters, 45(12), 6071-6080.

Gray, J. R., & Simões, F. J. (2008). Estimating sediment discharge. Sedimentation engineering–processes, measurements, modeling, and practice, manual, 110, 1067-1088.

Stokes, S., Bray, H. E., & Blum, M. D. (2001). Optical resetting in large drainage basins: tests of zeroing assumptions using single-aliquot procedures. Quaternary Science Reviews, 20(5-9), 879-885.

Wolman, M. G., & Miller, J. P. (1960). Magnitude and frequency of forces in geomorphic processes. The Journal of Geology, 68(1), 54-74.

—--------------------------------------------------------------------------------------------------------------

We thank the reviewer 2 (R2) for his/her constructive comments.Below, we provide replies to the main and some of the minor comments and detail how we will address them in a revised version of the manuscript.
—------------------------------------

The manuscript outlines a very nicely designed and conducted experiment that shows the effect of turbidity on the bleaching of sediments in shallow water environments (e.g., coastlines and rivers). The aim of the paper is clear and well-motivated by the exiting literature. The experiment is fairly well described and so are the results, which show how an increase in suspended sediment concentration leads to a stronger attenuation of the UV compared to the NIR wavelength, reflected in a positive correlation between residual doses and suspended sediment concentrations (largely caused by sediment resuspension and upwelling).

The study definitely has implications for a better understanding of luminescence-based sediment tracing in shallow water environments and, therefore, in my opinion, it deserves to be published. I have, however, some doubts mostly related to the motivation of certain aspects of the experiment set-up (see comments below).

**Main comments**

- You have chosen very high sediment concentrations for the bleaching experiment, which are typical of flood events. How do you justify this choice and not using other magnitudes representative of less extreme conditions as well? Did you expect a high turbidity threshold for the inversion? If so, you should mention this and explain on what grounds (e.g., observations from previous studies?).

(A) As R2 correctly assumes we not just expected a relatively high turbidity threshold but also that the effect of the differential bleaching will be more pronounced, i.e. better detectable, using high sediment concentrations.

- Why did you choose to allow light penetration only from the top? How do you think this might compare to natural light penetration? It would be interesting discuss this aspect alongside other comparisons you have made between the experiment set-up and natural settings.

(A) We blocked the light penetration from the sides because of several reasons. (1) In our setup with a circular flume, the grains are accelerated towards the container wall. Light penetrating from the sides could be regarded as simulating the floating of grains on the water surface making the turbidity less effective. (2) Much of the light entering from the sides will have been reflected from surfaces outside the flume, in our case mostly vegetation but also buildings. To avoid contamination of the spectrum by anthropogenic effects we shielded the sides of the container.

Because this point was also raised by R1 we will add another paragraph to the discussion that better justifies this setup.

- The water container used is relatively shallow compared to most natural settings. You mention that water depth might change light penetration, but it would be interesting if this aspect was discussed more in depth and also in relation to sediment bleaching.

(A) We agree that the effect of water depth on sediment bleaching could be discussed more thoroughly. Unfortunately our model setup is inappropriate to investigate the effect by ourselves. However there is at least one study that has shown that the OSL and IRSL signals are efficiently bleached in clear water down to 10 m depth (Rendell et al. 1994, *Underwater bleaching of signals from sediment grains: new experimental data*, QSR, 13). For the fluvial environment at least, water depth can thus be regarded as much less important for the sediment bleaching than turbidity.

We added to paragraph 4.3 the sentences:

*Concerning the effect of water depth on light penetration it has been shown that the OSL and IRSL signals are efficiently bleached in clear water down to 10 m depth (Rendell et al. 1994). For the fluvial environment at least, water depth can thus be regarded as much less important for the sediment bleaching than turbidity.*

- A lot of the discussion in relation to natural settings and references used are relative to fluvial environments but, as it is mentioned a couple of times through the manuscript, this study might have implications for other types of settings as well (e.g., shallow marine/coastal, glaciofluvial), so it would be nice to see throughout the manuscript references and examples relative to other types of environments as well. This would especially be coherent with the choice of using coastal sediments for the experiment.

(A) We agree with the reviewer and will add more references to coastal and lacustrine environments in a revised version of the manuscript.

We added references to Richardson (2001), Jacobs (2008); Alexanderson & Murray, (2012) and Ahmed et al., (2014) who focus on the coastal environment.

**Minor comments**

Lines 14-15: Sediment mixing due to advection?

Yes.

Line 36: I would say 'burial' rather than 'coverage'.

changed accordingly

Lines 48-66 and 334: The experiments have been performed on Miocene coastal sand. This is a problem which indeed affects shallow coastal systems as well (e.g., intertidal). I would mention the applicability of this study to other shallow water environments as well.

We added more references regarding applicability of this study to other shallow water environments in the introduction and the discussion.

Line 70: Is this threshold linked to your choice of concentrations? Did you have an assumption for what the threshold might have been?

(A) We hypothesized that the threshold is a function of turbidity/concentration. One could speculate that the threshold lies somewhere beyond 10 g/l, because this is the concentration, at which the irradiance curves from blue and red wavelengths cross over (see Fig. 3c).

Lines 81-83: Do you have any values for turbidity here from the literature? It could help justifying the chosen concentrations for the experiment.

(A) We will add more to the justification of our used concentrations because it was also asked by reviewer one. See response to R1s main comment.

Lines 83-84: Based on observations or assumed?

(A) Assumed.

Line 96: It might be good to explicitly state why you have chosen to use a saturated sample.

(A) Agreed.

Lines 97-107: I didn't quite understand why you introduce the collection of three samples but only describe the experiment conduction on one sample (GG1). If my understanding is correct and you have conducted all experiments on one sample, then I wouldn't mention the other two in the methods. If you have used GG2 and GG3 as well, then it might be helpful to state which samples you have used for what because I didn't manage to follow this properly through the methods section. If you have collected them for future experiments then you can mention that in the discussion within a 'future research/work' paragraph (roughly explaining what future research might focus on) with a supplementary figure showing their location, but I still wouldn't include them in the methods because it might be confusing for the reader.

(A) The reviewer is right and we will follow his suggestion.

Line 111: 'Tap' water.

corrected

Line 113: Paraphs include in brackets that the time intervals are stated below.

added a reference to Table 1

Lines 122-124: Why did you want to capture light only from the top? How do you think this might compare to natural light penetration? I would discuss this aspect later in the manuscript.

(A) We agree. See answer to major comments above.

Line 132: I assume here you have chosen shorter times because of more favourable bleaching conditions. Better to be explicit here as well.

(A) Agreed. See response to main comment above.

we added the sentence: *We chose shorter time intervals in the subaerial experiment due to the more favourable bleaching conditions.*

Lines 137-138 and 162-163: Do you have any references here?

We have added a reference to Reimann et al. (2015).

Line 165: 'Removed' should 'be remove'.

corrected

Line 186: 4) this also refers to Cs greater than 40, right?

(A) Yes.

Lines 207-218: It might be interesting to compare these changes also numerically, calculating the rate of decay.

Line 226: 'Was' expected.

corrected

Line 270: I wouldn't say 'turbidity is gone', I would rephrase this.

We rephrased this to "the turbidity is greatly reduced".

Lines 272-273: Unknown mineralogy with probable traces of organic matter? Have you not analyzed the sediment composition of the sample?

(A) We have analyzed the sample under the microscope but did not do x-ray diffraction to determine the very fine components. However, from the mineralogy of the bulk of the sample (99 % Qtz, 1% Fsp) we may assume that the fines simply reflect the same mineralogy. Because the sand originates directly from below a seam of brown coal, we expect organic material.

We will conduct further analysis to validate our assumptions and incorporate this into a revised manuscript.

We have analyzed the sample with XRD and EDX incorporated the results in the manuscript and the supplementary material.

Lines 274: How different? E.g., what if the samples are more organic?

(A) Here we refer to the grain size distribution of the sample, which can be different for identical SSCs. We will rephrase this sentence.

We could not verify the existence of organic material in the fine fraction of the sample.

Line 286: It would be interesting to discuss more in depth the water depth component and also in relation to bleaching.

(A) We agree. See response to major comment above.

Fig 1: 'Where should be were'.

corrected

Fig 7: It would be interesting to see how the $R^2$ compare between the two relationships.

We log-transformed the data and fitted linear functions. There is no clear trend visible when comparing the R2s between the two relationships. Thus we refrained from including this comparison in the manuscript.